# A Composite Pipeline for Forwarding Low-Latency Traffic in SDN Programmable Data Planes

Zhiyuan Ling [1,2], Xiao Chen [1,2] and Lei Song [1,2,*]

1 National Network New Media Engineering Research Center, Institute of Acoustics, Chinese Academy of Sciences, No. 21, North Fourth Ring Road, Haidian District, Beijing 100190, China
2 School of Electronic, Electrical and Communication Engineering, University of Chinese Academy of Sciences, No. 19(A), Yuquan Road, Shijingshan District, Beijing 100049, China
* Correspondence: songl@dsp.ac.cn

**Abstract:** With the rapid evolution of network technologies over recent years, emerging network services, especially industrial control networks, video conferencing, intelligent driving, and other scenarios, have put forward higher demand for the low-latency forwarding of network traffic. The existing flow caching and hardware acceleration methods only improve the overall forwarding performance of data-plane devices but cannot separate the forwarding process of low-latency traffic from others to reflect the priority of these flows. In this paper, we extend the POF southbound interface protocol and propose a marking method for low-latency flows, based on which we design a composite pipeline to achieve fast processing for low-latency traffic by introducing a fast-forwarding path. The experiments show that the fast path has a higher forwarding capability than the MAT pipeline in the POF Switch and can reduce the forwarding delay of low-latency flows by 62–68%. In a real network environment with a mixed traffic simulation, the reduction reaches 17–20% with no delay increment for the non-low-latency part.

**Keywords:** SDN switch; data-plane programming; fast packet processing

## 1. Introduction

In recent years, networking technology has been rapidly developing within a variety of application areas. Fields such as industrial control networks, Information-Centric Networking (ICN), and Data Center Networking (DCN) constantly update the requirements for network throughput and latency. Services such as industrial control and unmanned driving are sensitive to network latency and require high reliability. The transmission of emergency information such as vehicle status requires millisecond latency. The communication latency between devices in industrial networks also needs to be controlled within 10 ms [1–3].

DCNs have become an important part of the Internet infrastructure, providing Internet users with reliable network services. DCNs are internally interconnected by a huge number of servers and place high demands on the communication latency and bandwidth between the servers. For example, the end-to-end latency between servers often needs to be on a microsecond level [4].

In recent years, data center service traffic has been growing exponentially, and data center expansion is trending rapidly. The year 2020 saw a $10\times$ increase in total servers and a $50\times$ increase in global data [5]. With increasing service complexity, these issues bring many challenges for cloud DCNs, and software-defined networking (SDN) becomes the key to building cloud data centers [6]. DCNs in the cloud computing era have put forward many new demands on the network. The first problem to address is the automation and centralized control of large-scale networks, and some other emerging IT application architectures also require the network to be more intelligent. Traditional networks have long struggled to meet these demands, but SDN architecture can fit these needs perfectly [7].

Network flows have their QoS (Quality of Service) requirements, among which low-latency transmission is a common and significant attribute. In scenarios such as industrial control networks and telemedicine, the low transmission latency of control flows is a crucial guarantee of the network QoS. In addition, some traffic has low-latency forwarding requirements but without evident flags in the packet headers, such as machine-class communication between servers in DCNs. In DCNs, information such as the addresses of the devices is usually fixed for a long time, and the routing or other related operations of the communication data are also relatively stable. These communication flows can be transmitted in a fast path due to stable routing. We call the network flows described above low-latency flows.

It is clear that in a single pipeline based on multi-level MATs (match-action tables), low-latency flows are processed in the same path as others. As network services are diversifying, the pipeline processing flow will also become complicated, ultimately increasing the processing latency of network flows. Although some programmable switches use methods such as flow caching [8] to reduce the forwarding latency of some flows, flow caching does not distinguish low-latency flows from others, so it cannot solve the above problem.

Our approach upgrades the multi-MAT process to a composite pipeline which processes low-latency flows in a more efficient path. We borrow the idea of flow caching in the path and optimize it to reduce the time and space complexity of the scheme through hierarchical caching. In addition, we use a low-latency traffic identification method that combines the control plane and the data plane to restrict the cached forwarding rules and extract the low-latency traffic for separate and efficient processing. The contributions of this paper can be summarized as follows.

- We extend the POF (Protocol-Oblivious Forwarding) southbound interface protocol, based on which we provide a method to label low-latency flows.
- We design a fast-forwarding path (FFP) that prioritizes low-latency flows without affecting the work of the MAT pipeline. The design includes the method to extract forwarding rules from MATs, the process of forwarding low-latency traffic, real-time FFP updating, etc.
- We implement the above scheme on a DPDK-based POF Switch and compare it with the original POF Switch. The results show that our proposed approach can effectively reduce the forwarding latency of low-latency traffic.

The structure of this paper is organized as follows. In Section 2, we show the recent work in the field. In Section 3, we propose the FFP design. Section 4 shows our experimental results. Finally, we conclude the whole paper and discuss our future work plans.

## 2. Literature Review and Background Work

### 2.1. Background Work

As a pioneer of SDN, OpenFlow [9] puts forward the packet processing pipeline based on MATs, which has become a common structure for subsequent SDN data planes [7]. A prepositive parser installed in the OpenFlow switch parses packets according to the protocol fields defined in the OpenFlow Switch Specification [10]. Due to the finiteness of the defined fields, the programmability of OpenFlow switches is limited when processing packets with emerging network protocols.

The POF [11] and P4 [12] were proposed one after another to achieve higher data-plane programmability. P4 is a high-level language for SDN data-plane programming to configure the packet processing logic of the switches. The protocol-independent switch architecture (PISA) [13] is a pipeline engine that supports programmable packet processing for P4 data planes. A programmable parser, multi-stage programmable MATs, and a programmable deparser form a pipeline for P4 data-plane programming. The programmable parser provides support for the processing of emerging network protocols. In the OpenFlow and P4 switches, the front parser is a mandatory module for all flows. During the parsing, the parser does not know which fields are needed for subsequent MAT queries, so it parses as



many protocol fields as possible. However, most of the packet processing uses only a small fraction of the protocol fields, so a large part of the parser's work is redundant.

The POF is a protocol-independent southbound interface protocol that builds on OpenFlow. The POF defines arbitrary packet fields or other status data in {type, offset, length}, which provides greater programming flexibility than OpenFlow [14]. Thanks to this design, POF switches do not need a front parser to parse packets. The task is left to the control plane instead. Control-plane applications load appropriate flow tables and table entries into data-plane devices based on their respective requirements. When packets are processed in the pipeline, only the data required for the query need to be extracted based on the matching fields in the current table before searching the entries. This approach of real-time parsing during the table lookup is more flexible than the P4 and OpenFlow switches with front parsers. The parsers are generated in the configuration phase and cannot be updated at the data-plane runtime, while the runtime flow table updating in the POF switches solves this problem.

At the cost of enhancing the data-plane programmability, parsing packet headers hierarchically in MATs increases the packet processing latency and creates difficulties for the flow cache design [15]. Flow caching uses the entire packet header or parsed fields as the input, but some of the fields may not be of interest in the pipeline, which can make the hash results very scattered and thus reduces the cache hit ratio. An often-studied approach is rule-extraction and aggregation [16–21]. The related research integrates the header fields used in each phase of the packet processing into a complete forwarding rule. However, for the multi-MAT pipeline in the POF Switch, the time complexity of the field integration is quite large. The result of the matching field permutation grows exponentially with the length of the pipeline, which puts pressure on the storage space.

### 2.2. Related Work

### 2.2.1. Global Routing Adjustment

Some studies were carried out in the field of fast packet processing. According to [22], rule placement optimization improves the packet forwarding performance of DCNs. The method obtains information about the entire network and analyzes all existing traffic. Then, a suitable forwarding rule placement policy can be found to improve the traffic processing capacity. However, these optimization policies are usually static. It is unaffordable to update the rules in all switches immediately when the status of the traffic, such as the network address, changes.

### 2.2.2. Flow Caching

One of the core components of traffic forwarding is packet classification. Ternary Content Addressable Memory (TCAM) uses a flexible wildcards configuration to classify packets efficiently at the hardware level. However, the cost and power consumption of the TCAM limit the number of classification rules in data-plane devices, which exemplifies the importance of the software implementation of packet classification algorithms. The optimization of the algorithms is still hard due to the average CPU cache level. Developers have focused their attention on a proven solution, flow caching. Rule placement optimization treats the rule space as a priority resource, while flow caching policies efficiently use space to store the latest used forwarding rules. Compared to rule placement optimization, rule caching is a superior approach that allows traffic-based control to provide both a high performance and scalability, especially in large-scale DCNs.

Open vSwitch (OVS) is a classic commercial OpenFlow switch, and it implements an exact-matching flow cache at the first version [23]. The device performs successive classification steps in a slow path for the first packet in a flow and caches the exact-matching results. Subsequent packets are then processed in the fast cache module by a hash lookup, with the key of the hash based on the whole packet header. In the slow path, the worst-case lookup time for an MAT is $O(N)$, where *n* is the number of rules, while the time complexity of the hash table in the cache is $O(1)$. Similar mechanisms are common on x86-based

devices. Although flow caching is a significant improvement, single-connection caching still requires the slow path to be involved in each new transport connection, even if the resulting forwarding decisions are similar across multiple connections. For example, when using exact matching to look up the flow cache, traffic from different source ports requires separate cache entries.

As a further improvement, we can restrict the range of the matching fields in flow cache entries to no more than L2 and L3 headers. As an example, the OVS implements MegaFlow and reverts to a connection cache based on exact matching only when the forwarding decision on the slow path depends on the L4 header. As long as the matching is based only on L2 and L3 headers, the computed cache entries do not need to be updated for each transported connection, and the fast path can process new connections without sending packets to the slow path. However, forwarding decisions increasingly depend on the L4 headers, on which almost all network services work. For instance, firewall services filter out all traffic beyond the SMTP, HTTP traffic needs to be redirected to a caching proxy, etc. Thus exact-matching-based connection flow caching becomes the only practical flow caching option.

### 2.2.3. Hardware-Based Acceleration

Several studies improved the performance of software packet-switching solutions by leveraging hardware-based acceleration. PacketShader [24], a software router, is developed based on an acceleration framework based on a GPU. PacketShader implements an I/O engine for fast and efficient packet processing. Some functions, such as routing table lookup and IPsec encryption, are offloaded from the main processor to the GPU. The I/O engine works for kernel-level packet processing operations, and other packet processing operations are performed in multi-threaded applications in the user space. Based on the above design, PacketShader can achieve a high throughput of 40 Gbps.

A hybrid architecture switch is designed with a similar method. A network processor (NP) is programmed to execute tasks, such as receiving/sending packets, processing packets, managing queues, etc. The software on the host and the NP acceleration card make up the implementation based on the above design. The OpenFlow software in the host uses the kernel module to communicate with the NP acceleration card via the PCIe bus. The packet latency is reduced by 20% compared to traditional OpenFlow switches.

The comparison of the approaches introduced in the above literature and our scheme in this article is listed in Table 1.

**Table 1.** Summary of approaches in the literature and our scheme.

| Technology | Reference | Contributions | Shortcoming |
|---|---|---|---|
| Global routing adjustment | Rule replacement optimization [22] | Improved forwarding performance | High time complexity |
| Flow caching | Open vSwitch [23] | High performance and scalability | Low cache hit ratio for single-connection traffic |
| Hardware-based acceleration | PacketShader [24], OpenFlow switch with NP acceleration card [25] | Higher forwarding performance than software solutions | Less programmability |
| Hierarchical-hash rule caching | This article | Extract and fast-forward low-latency traffic | Little latency increasement for normal traffic in some scenarios |

## 3. Problem Description

The conducted studies have focused their efforts on the improvement in the overall forwarding capacity of the data plane. These studies have indeed contributed a lot to the performance optimization of data-plane devices, such as switches; however, these

approaches take no care of the traffic types, resulting in traffic with special QoS requirements such as time-sensitivity ones not being identified and optimized separately in the processing flow. To overcome these difficulties, our study needs to address the following questions:

- How to identify and extract low-latency traffic? The POF Switch does not have a packet parser module to obtain the packet header protocol fields by parsing the packets in advance, which makes the design of the front-end traffic identification module very difficult. Therefore, we will use a runtime identification scheme to jointly participate in the identification and extraction of low-latency traffic by adding tags to the packets and forwarding rules in combination with control-plane decisions.

- After extracting the low-latency traffic, how to design a dedicated processing path with a better forwarding performance than the original MAT pipeline? Due to the nature of the POF, methods such as full-domain hashing that are used in other switch solutions for flow caching implementation would introduce an excessive time and space complexity, so we consider following the structure of a hierarchical processing pipeline. This requires us to improve the execution efficiency of individual processing stages. Further, we also need to optimize the lookup algorithm for certain tables for different match field types.

### 4. The Proposed Mechanisms

In this section, we propose the data-plane implementation based on a software composite pipeline architecture. We design an FFP which runs parallel to a multi-MAT pipeline, and we achieve fast processing of low-latency flows through the interaction between the fast path and the pipeline. Figure 1 shows the network system that our work applies to, and the contributions focus on the POF Switch and the ONOS controller.

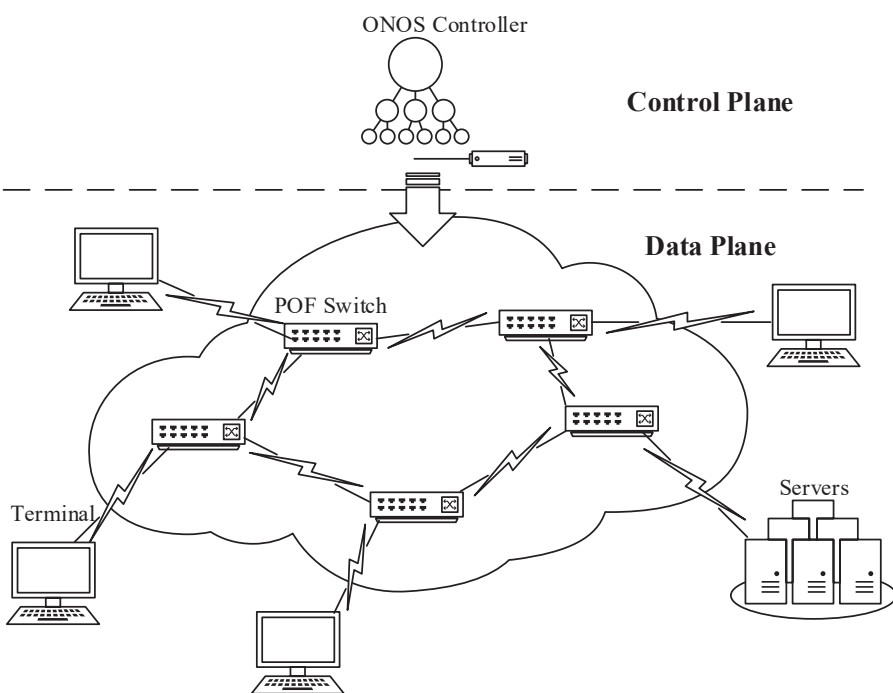

**Figure 1.** The network system model where our approach works.

### 4.1. Definition and Recognition of Low-Latency Flows

As we mentioned earlier, low-latency flows, in this paper, include traffic with requirements of low forwarding delay, traffic with relatively fixed forwarding rules, etc. For the former, data-plane devices, especially protocol-oblivious white-box switches such as POF Switch, are insensitive to the message content and cannot identify low-latency flows with the information in messages. In SDN, however, there are no relevant restrictions on the

control plane, and control-plane network applications understand the semantics of the various fields in messages. The QoS attributes of traffic are usually explicitly represented in the message by certain fields, so the control plane can easily generate appropriate forwarding rules for low-latency traffic based on these fields and load them into data-plane devices. In other words, the control plane can determine whether a forwarding rule serves low-latency traffic based on the content of the rule. These forwarding rules are stored in the data-plane device as MAT entries, so we extend the existing POF southbound interface protocol. The new protocol agrees with the control plane to add a low-latency flag to an entry when placing it, to identify whether the entry is involved in the forwarding of low-latency flows, as shown in Figure 2. For traffic with relatively fixed forwarding rules, there is no obvious field in the message describing the low-latency property, but these rules are usually permanent MAT entries or are set with a large timeout. Based on this premise, the data-plane device assigns low-latency properties to MAT entries and packets at runtime by analyzing the packet's match records in the pipeline and the hit statistics of the table entries. That is, the low-latency flags of these MAT entries are initialized to *FALSE* when they are sent down to the data plane and changed in real time as needed. We refer to an MAT entry with the low-latency flag set *TRUE* as a low-latency entry or low-latency traffic.

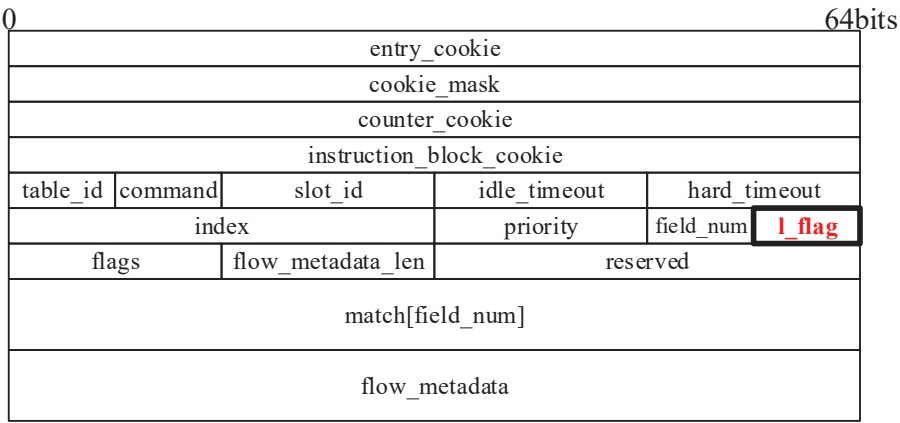

**Figure 2.** FLOW_MOD message of the extended POF southbound interface protocol.

The fact that a packet hits a low-latency entry in the pipeline cannot prove that the packet belongs to a low-latency flow, because packets of other flows may also hit low-latency entries. Because a flow in the switch is defined by entries in multi-level flow tables together, the packet can be considered to belong to low-latency traffic only if all the hit entries during the processing are low latency. To identify low-latency traffic, an easy method is to actively integrate low-latency entries. By arranging and combining these entries in each level of the MATs to form a forwarding rule table (FRT) with an overlong matching field, then each low-latency flow must hit an appropriate forwarding rule in the FRT. However, this approach has some drawbacks. Firstly, the size of the generated FRT would be unacceptable because rule consolidation requires a Cartesian product of the low-latency entries from each level. As the number and length of pipeline branches grow, the number of consolidated forwarding rules will eventually explode. In addition, only a small fraction of the rules can be hit by low-latency flows, making the use of resources inefficient. Secondly, not all packets hitting the rules in the FRT belong to low-latency traffic, and the hit rules may not be the optimal forwarding policy, because the packet may match higher priority entries in the pipeline, which will lead to "false caching". This problem was investigated in some studies, such as CacheFlow. However, the algorithm complexity grows with the length of matching fields in the FRT, which makes the performance not as well as expected.

To solve the above problem, our proposed approach marks the packets in addition to the MAT entries, which requires the data plane to support state programmability. In a DPDK-based POF Switch, the packet states are stored in the packet metadata, where we

can set a flag to mark the current low-latency state of the packet, as shown in Figure 3. When a packet enters the switch, this flag is initialized to *TRUE*, indicating that the packet may belong to a low-latency flow up to now. During the subsequent processing, the flag will be corrected to *FALSE* with the result of the MAT lookup.

When the packet hits a normal entry and the packet flag is *TRUE*, the packet is no longer part of a low-latency flow, and the flag needs to be updated to *FALSE*. In other cases, the flag remains unchanged.

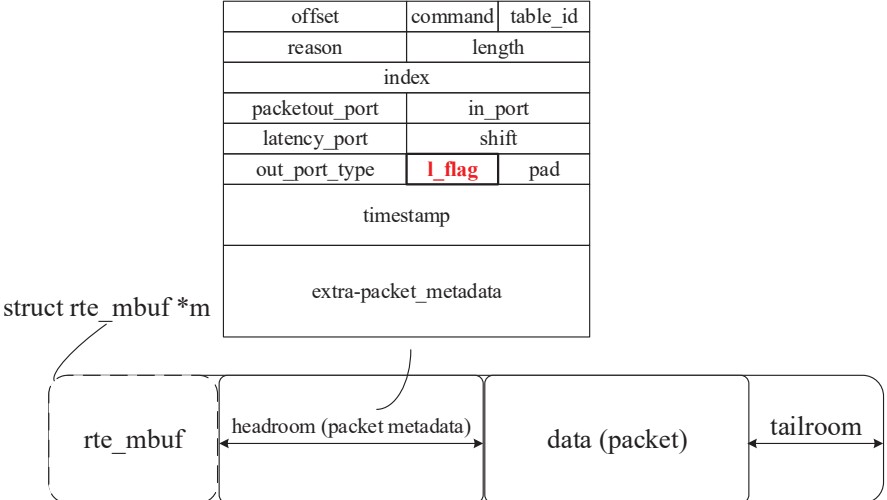

**Figure 3.** The structure of *rte_mbuf* which stores packets and the packet metadata.

*4.2. Composite Pipeline Architecture*

In data planes with a generic multi-MAT pipeline, low-latency flows are mixed with others and are forwarded via the same channel. Such an undifferentiated processing path design can hardly guarantee the fast processing of flows with the need for low-latency forwarding. To solve this problem, we propose a composite pipeline structure based on an FFP, which splits low-latency flows from ordinary ones and spends less time forwarding them.

4.2.1. Architecture Design

As mentioned earlier, whether a flow is low latency depends on both the packets and the hit MAT entries, so we cannot identify and distinguish the traffic before it enters the pipeline. The recognition performs during the forwarding process. To be specific, ordinary flows are processed along with low-latency ones at the beginning and are gradually filtered out and sent to the pipeline for further processing. This idea is borrowed from the flow caching of OVS. In OVS, all traffic entering the switch first goes to MicroFlow and MegaFlow for cache lookup. Some flows hit the cache and are forwarded foremost, and the rest are sent to the slow pipeline to re-query the MATs. Unlike OVS, in our scheme, after moving from the FFP to the pipeline, the ordinary flows have no need to match from the first MAT but start processing at the intermediate stage, i.e., the processing of ordinary flows partly gains the benefit of FFP acceleration. This design partially offsets the increased latency due to packet transfer between modules and additional query steps. The data plane with the FFP is shown in Figure 4.

The FFP is a pipeline-like structure that consists of multiple cascaded lookup tables (LUTs). Each LUT corresponds to an MAT in the main pipeline and stores the hit records of low-latency rules. As can be seen, the approach proposed in this paper differs from common flow caching schemes. A flow cache is usually set in front of the main pipeline and uses the entire packet header as the input for hash lookup, which is also called the global-field hash. Obviously, this approach is unfriendly to short-connected caches. Any slight change in packet header fields will totally change the hash value, which results in a

poor hit rate of the cache. From a global perspective, the average traffic forwarding latency increases because of the prolonged forwarding process.

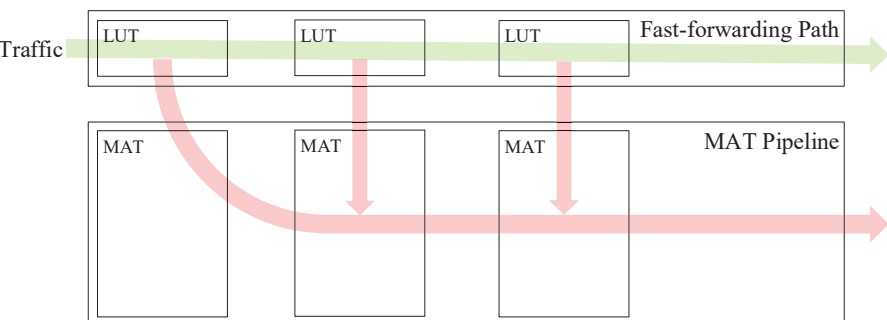

**Figure 4.** The architecture of the composite pipeline with an FFP.

Our proposed scheme uses the composite structure shown in Figure 4. Each LUT in the FFP is composed of low-latency entries extracted from the corresponding MAT in the pipeline. Packets entering the switch are first processed in the FFP. If a packet does not hit any low-latency entry in an LUT, the packet will be sent to the pipeline to look up the corresponding MAT and continue its subsequent forwarding process. It is easy to see that with the introduction of the FFP, the length of the forwarding process is almost the same as the pipeline-only scheme, so we need to start our research work from the perspective of optimizing the single LUT in the FFP.

The MATs in the pipeline are designed based on the principle of protocol hierarchy, so the matching fields of an MAT only cover fields of a certain protocol in the packet header. For this characteristic, we borrow the idea of flow caching and use hash tables as the basic structure of LUTs in the FFP. Our approach alleviates the problems of time and space complexity to some extent by hierarchical hash. We narrow down the idea of the global-field hash flow caching to the scope of a single MAT. The matching fields of one MAT are much fewer and the length is shorter as well, which is more suitable as a hash input and will get higher space usage efficiency in the limited hash table space. Furthermore, it is unnecessary to reconstruct a hash table for each MAT. Some types of MATs can split flows efficiently enough, so adding a hash table will bring extra processing latency for non-low-latency traffic.

In MAT-based data planes, MATs can be divided into the following categories according to matching methods: linear, exact matching (EM), longest prefix matching (LPM), and mask matching (MM). The entries in linear tables have no matching fields but only an index, an instruction block, timers, counters, etc. The packet to be processed will carry an entry index in its metadata when it executes "GOTO_DIRECT" in the previous MAT. The corresponding instruction block will be executed directly according to the index in the linear table, so the searching time complexity of a linear table is $O(1)$. EM tables are implemented using hash tables, so the searching time complexity is also $O(1)$. LPM is a common algorithm used by routers in IP networks to select entries from routing tables. The classical LPM algorithm is implemented using a binary tree, and the searching time complexity is $O(logL)$, where $L$ is the length of the matching field (IP address). DPDK LPM library provides optimized LPM algorithms, respectively, for IPv4 and IPv6, which reduces the searching time complexity to $O(1)$ through multi-level hash tables. In future networking, we should not limit the application scenario of LPM to IP address searching. Other situations, such as hierarchical naming lookup in named data networking (NDN), also need LPM. MM tables are used in the case of arbitrary matching, such as 5-tuple searching in access control lists (ACL). MM sets the effectiveness of each bit in the match field by a mask to search arbitrary packet header fields. DPDK provides an ACL algorithm for MM tables based on trie, and the searching time complexity is $O(L)$.

For EM tables, the searching time complexity is already $O(1)$, and there is almost no room for further optimization, so we do not need to set another LUT in the FFP for these MATs. In other words, the pipeline can share EM tables with the FFP to process packets together. According to this design, any packets in these two types of MATs can complete their current stage of processing. A packet will directly go to the next-level stage whether it hits a low-latency entry or a normal one. For other types of MATs, it is necessary to set hash-based LUTs in the FFP to reduce the searching time complexity.

### 4.2.2. Preliminary Benefit Analysis

For the above design of the FFP, we can further refine each part and estimate the benefit of forwarding latency. We can first consider the case of a single MAT. Because we do not create extra LUTs in the FFP for linear tables or EM tables, the gain of the two types of tables is 0. Therefore, here we mainly discuss the case of other classes of MATs. Assume the length of the matching fields in the MAT is $l$ and the size is $n$. The searching time of the MAT can be considered a function on $l$ and $n$, $t_{pipeline}(l, n)$. The expression of this function varies for different MATs as shown in Table 2:

**Table 2.** Searching time of different MATs.

| Table Types | Linear | EM | MM | LPM |
|:---:|:---:|:---:|:---:|:---:|
| **Searching time** $t_{pipeline}(l, n)$ | $C$ | $k_{hash}l + C$ | $k_{trie}l + C$ | $k_{bin-tree}l + C$ |

In addition, due to the hash-based implementation, for the LUTs in the FFP, the searching time can be unified as

$$t_{fp}(l) = k_{hash}l + C \tag{1}$$

Based on the above definition of searching time, we can obtain the searching time gain of the FFP in a single-level LUT:

$$W_d(l, n) = t_{pipeline}(l, n) - t_{fp}(l) = \begin{cases} k_{hash}l - nt_{cmp}(l) + C, & \text{linear tables} \\ (k_{type} - k_{hash})l + C, & \text{others} \end{cases} \tag{2}$$

Up until now, we consider only the gain of the forwarding latency for low-latency flows. Others miss in the FFP and are sent to the pipeline for further searching, which will give a negative gain in the processing time:

$$W_d(l, n) = -t_{fp}(l) - t_{transfer} \tag{3}$$

$t_{transfer}$ stands for the time cost by transferring the packets from the FFP to the pipeline.

### 4.2.3. Runtime Updating of the FFP

After the architecture design, we initially determine the basic composite pipeline structure. The next task is to extract the low-latency entries from the MATs in the pipeline into the FFP. Low-latency flows are identified by the low-latency flags in both MAT entries and packet_metadata, which makes the FFP updating a packet-driven reactive process. Figure 5 depicts the specific steps of the updating.

Similar to flow caching, the FFP is positioned in the switch as the entrance for traffic processing. During the initialization phase, every LUT in the FFP is empty. As shown in Figure 5, packet *Pkt*1 enters the FFP and directly skips the first empty LUT and is then forwarded to the MAT in the pipeline for processing. *Pkt*1 hits a low-latency entry *r*3 in the MAT. Because the low-latency flag of *Pkt*1 is initialized to *TRUE*, the flag does not need to be changed and *Pkt*1 retains the possibility of belonging to a low-latency flow. After *r*3 is hit, it needs to be extracted and updated into the FFP. Before updating, we need to perform a restricted-field hash of *Pkt*1: the input to the hash is a contiguous segment of data that

originates from the parts of the packet that are covered by the MAT matching fields. For example, table *T*0 divides packets by their input port, destination MAC, and network-layer protocol type. This piece of data was extracted before searching the MAT, so we do not need to re-extract it. Through the hash calculation, we obtain a key, *Key*1, which is used as an index for LUT searching and modification in the FFP, so we need to record *Key*1. To simplify the description, we bind the key to the entry *r*3 in the form of a linked list. Then, we populate *r*3 to the corresponding position in the LUT according to the obtained key. If the packet matches a non-low-latency MAT entry *r*5, the low-latency flag of the packet should be updated to *FALSE*, indicating that the packet is no longer likely to belong to low-latency traffic, and the subsequent processing of the packet can only be completed in the pipeline.

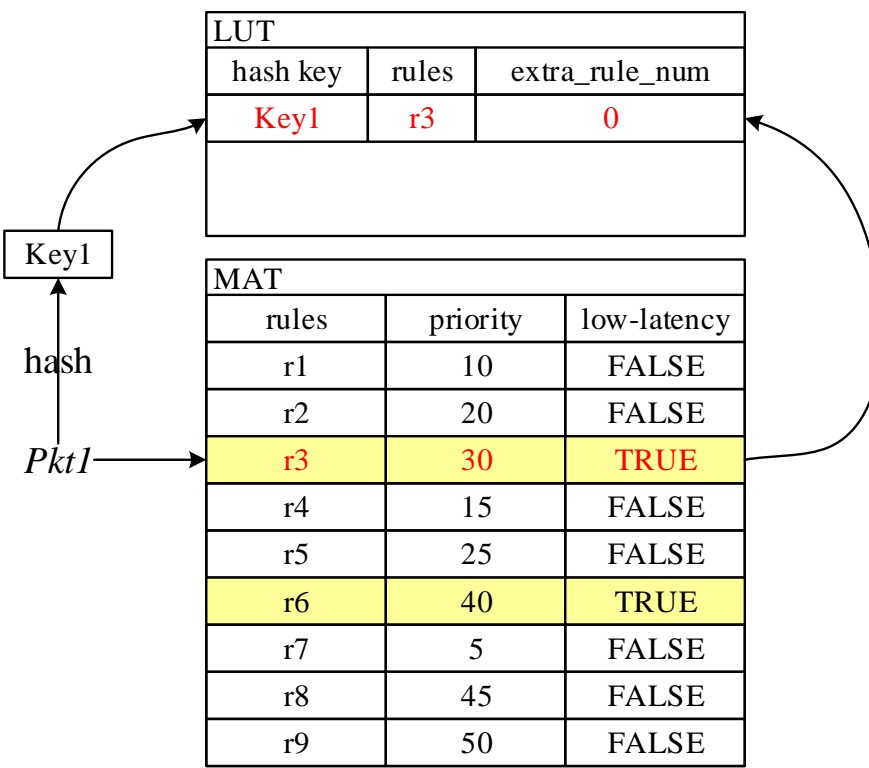

**Figure 5.** The updating progress of LUTs in the FFP.

After the above processing, subsequent packets belonging to the same flow as *Pkt*1 enter the switch and will directly hit entry *r*3 in the FFP, avoiding searching the original MAT. However, this is an ideal situation built on the condition that the original MAT has not changed. Some new entries with higher priority than that of *r*3 may be added to the MAT. In this case, we need to update the LUT in time. Figure 6 shows an example of updating.

Three new entries, *r*10, *r*11, and *r*12, are added to the MAT *T*0, all of which have higher priority than *r*3. We need to screen these new rules during the updating phase of the MAT, eliminating those that conflict with the low-latency entries recorded in the FFP—in this case, only *r*3—and selecting only those that have dependencies on *r*3 for the next step.

The matching fields of all MAT entries can be represented by value, mask, where the mask is used to identify which bits in the value are valid. For different types of MATs, the arrangement of bit-1 in the mask differs. For example, LPM rules only match prefixes, so the form of the mask is as follows:

$$\overbrace{11\ldots1}^{n}\overbrace{00\ldots0}^{L-n} \tag{4}$$

*n* is the prefix length. For the entries in MM tables, the form of the mask is not required, so MM tables can be used to match arbitrary fields. We can first consider the computation of rule dependencies in MM tables because this is the most general scenario. The dependencies between MAT entries can be obtained by intersection, and the specific calculation steps are as follows:

$$common\_mask = mask1\&mask2 \tag{5}$$

$$result = cmp\_neq(value1, value2, common\_mask) \tag{6}$$

*common_mask* is the intersection of the rule masks and represents the bits that the two rules both care about. Then, we compare the common bits from each rule: if there are any unequal bits, the two rules conflict; otherwise, the packet space covered by both has an intersection.

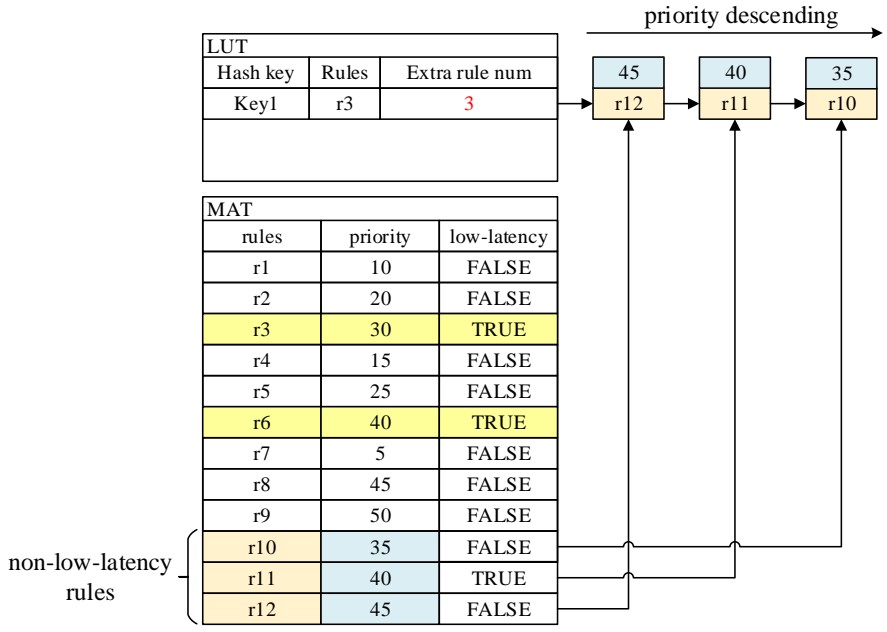

**Figure 6.** LUT updating when non-low-latency rules are added in the MAT.

To avoid cache mistakes when packets are processed in the FFP, we need to verify the correctness of the search result. From Figure 5, we can see that the LUT entry structure in the FFP has a counter, *extra_rule_num*, and a linked list, *extra_rules*. The ordinary MAT entries having dependencies on the recorded low-latency rule are stored in the linked list in descending order of priority. *extra_rule_num* indicates the length of the current linked list. When the packet matches a low-latency rule recorded in the FFP and *extra_rule_num* is greater than 0, the extra rules need to be traversed to verify the correctness of the match result. If a normal rule in the linked list is matched, then the low-latency flag of the packet is corrected to *FALSE*, and the packet will be sent to the next MAT in the pipeline for further processing.

The cost of traversing the linked list varies with its length. Therefore, when we update the linked list, we need to calculate the average time required for traversal. We calculate a weighted value according to the priority of each rule. In addition, when an MAT is created in the pipeline, we calculate its searching time immediately. The factors that affect the searching time of an MAT are none other than the search algorithm, the length of the matching fields, and the table size, which are immutable once the MAT is built. Thus, we can calculate the search time of the MATs in advance. When the correctness of the FFP search result needs to be affirmed, the solution with the shorter time will be chosen based on the search time calculated earlier.

If the packet hits another rule during the verification, the previously hit low-latency entry record must be invalidated, i.e., the record must be deleted in the FFP to avoid subsequent packets from hitting again. If the packet does not hit a new rule, it means that the record is still valid and can be directly used as the matching result for subsequent packets, so the linked list, *extra_rules*, can be cleared and the corresponding counter, *extra_rule_num*, will be reset 0.

We propose an optimized matching correctness verification scheme for LPM tables. The scheme simplifies the above universal verification process by taking advantage of the specificity of the matching field form of the LPM tables and the binary-tree search structure. Rules in the LPM tables have no priority, and if packets can match multiple rules at the same time, only the rule with the longest prefix will be used as the result. In another word, the prefix of an LPM entry plays a similar role to the priorities in other MATs and is used to determine the final matching result of the packet. As we mentioned earlier, for new MAT entries, we only need to focus on those with higher priority than recorded low-latency rules in the FFP. Therefore, in LPM tables, we only need to consider the new rules with longer prefix lengths.

The intersection relationship between MAT entries is intuitively represented based on the special lookup structure of LPM tables, which is a binary tree with each table entry distributed among the tree nodes, including all leaf nodes and some intermediate nodes. If a new entry intersects with the packet space covered by a low-latency rule recorded in the FFP, the nodes corresponding to them must be on the same path from the root node to some leaf node. Because we only consider new rules with longer prefixes, if two rules have an intersection, then the node of the new rule must be a descendant of the low-latency one. We already know this result when updating the binary tree, because the insertion process of the new entry node will go through the node of the low-latency rule. Therefore, we do not need to record these new rules in the form of a linked list in the FFP. Instead, we only need a flag, *verify_flag*, to identify whether we need to continue traversing the binary tree as well as the address of the hit rule node, as shown in Figure 7.

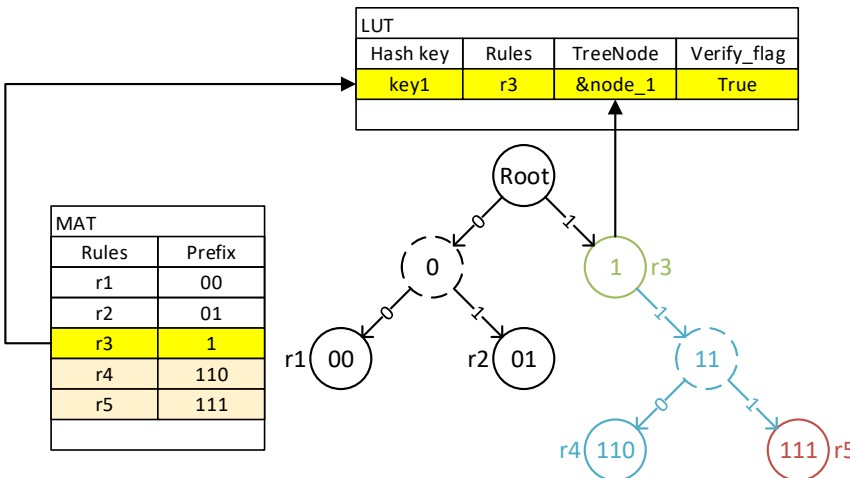

**Figure 7.** LUT design for LPM tables.

When a packet hits a low-latency rule in an FFP LUT, *verify_flag* should be checked. If a further search is required, the packet starts searching the subtree from the provided node address. If another entry is matched in the subtree, the previously hit low-latency rule record should be deleted from the FFP. Otherwise, *verify_flag* resets, and subsequent packets no longer need verification.

The above discussion is conducted for normal-rule addition scenarios. For new low-latency MAT entries, the conclusion differs a bit in the process. If a packet hits a new low-

latency rule during the verification, the previously hit record in the FFP will not be deleted but modified with the latest hit one to update the processing logic of subsequent packets.

In Section 4.2.2, we performed benefit analysis in the ideal scenario where each low-latency flow hits a rule in the FFP. However, we need to take the matching correctness verification into consideration. After the above analysis, the time required for matching correctness verification can be described as follows:

$$T_v = min\{t_{list}, t_{pipeline}\} \tag{7}$$

$t\_list$ and $t\_pipeline$ are, respectively, the time required to traverse the linked list *extra_rules* and search the MAT in the pipeline. We estimate $t\_list$ using a priority-weighted average:

$$T_{list} = p't_{cmp} = \frac{1}{N}(\sum_{i=1}^{N} ip_i)t_{cmp}(l) \tag{8}$$

$N$ is the length of *extra_rules*, and $p_i$ is the priority of each extra-rule. $t\_cmp(l)$ represents the time required to execute a comparison.

For all low-latency traffic entering the LUT, assuming that the percentage of traffic that directly hits a rule without verifying the matching correctness is $\rho$, the average processing time of all low-latency flows can be denoted as:

$$
\begin{aligned}
T_l &= \rho t_{fp} + (1-\rho)(t_{fp} + t_{verification}) \\
&= t_{fp} + (1-\rho)t_{verification} \\
&= k_{hash}l + (1-\rho)\min\{\frac{1}{N}(\sum_{i=1}^{N} ip_i)t_{cmp}(l), t_{pipeline}(l,n)\} + C
\end{aligned} \tag{9}
$$

It is easy to see from the above equation that the average processing time of low-latency flows is mainly affected by $\rho$. In practical situations, only the first packet in a flow needs to be verified for correct matching during processing, unless there are frequent rule updates at the same time. However, from experience, rule updates are similar to network flows on the temporal dimension, usually having aggregation and burstiness. Therefore, $\rho$ can maintain relatively large values in the vast majority of cases, and the average processing time of low-latency flows can be approximated as the same as the searching time of a hash table.

In the previous section, we only consider traffic missing in the FFP when searching a hash table to calculate the average processing time of ordinary flows. However, part of normal traffic may match low-latency rules in the FFP and be founded not low-latency after the match correctness verification. Assuming that the query misses account for $\rho'$ of all normal traffic, the ratio of ordinary traffic filtered out during the validation process is $1 - \rho'$. Then, the average processing time of ordinary flows can be denoted by:

$$
\begin{aligned}
T_n &= \rho'(t_{fp} + t_{pipeline}) + (1-\rho')(t_{fp} + t_{verification}) \\
&= t_{fp} + \rho't_{pipeline}(l,n) + (1-\rho')t_{verification} \\
&= k_{hash}l + \rho't_{pipeline}(l,n) + (1-\rho')\min\{\frac{1}{N}(\sum_{i=1}^{N} ip_i)t_{cmp}(l), t_{pipeline}(l,n)\} + C
\end{aligned} \tag{10}
$$

Similar to low-latency traffic, the packet filtered out during the verification is usually the first packet in a normal flow, and the rule updates also have some effect on the value of $\rho'$. For the same reason, we can approximate the above equation as $k_{hash}l + t\_pipeline(l,n)$. In other words, after the introduction of FFP, the processing of ordinary traffic has an additional hash table lookup phase compared to the original one.

From the results of the above analysis of single-stage processing, it seems that the gain from introducing an FFP is not significant enough. However, ultimately, we have to analyze the whole problem from the perspective of a multi-stage pipeline.

Figure 8 briefly depicts the forwarding process of traffic within the switch, where $P_i$ denotes the MAT at level $i$ in the pipeline, and $L_i$ is the LUT in the FFP corresponding to $P_i$. The cases except EM tables are discussed here first. In the figure, $\rho_i$ indicates the proportion of traffic that hits low-latency rules and goes to $L_{i+1}$ among the traffic processed by $L_i$. $\rho_0$ is initialized to 0. From a simple analysis, we can obtain the percentage of fast-processed and slow-processed flows to the total traffic:

$$R_{L_i} = \prod_{j=0}^{i} \rho_j \tag{11}$$

$$R_{P_i} = R_{L_{i-1}} - R_{L_i} \tag{12}$$

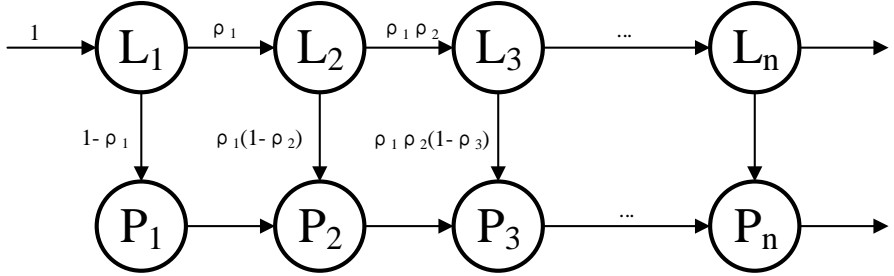

**Figure 8.** Traffic forwarding process model.

The low-latency traffic handled entirely by the FFP represents only a fraction of the total traffic, and the specific ratio is $R_{L_N}$. All other traffic undergoes a portion of fast-path processing, more or less. Assuming that the traffic size (the number of packets) over some time is $F$, the number of packets for low-latency traffic is

$$F_L = FR_{L_N} \tag{13}$$

and the one for other flows is

$$F_N = F(1 - R_{L_N}) \tag{14}$$

We denote the average time for node $L_i$ and node $P_i$ to process a packet as $t_{l_i}$ and $t_{p_i}$. Then, the cost time of each node in the FFP when the switch processes the $F$ packets is

$$T_{L_i} = FR_{L_{i-1}} t_{l_i} \tag{15}$$

So, the processing time of the whole fast-processing path is

$$T_L = \sum_{i-1}^{N} T_{L_i} = F \sum_{i=1}^{N} R_{L_{i-1}} t_{l_i} = F \sum_{i=1}^{N} (\prod_{j=0}^{i-1} \rho_j) t_{l_i} \tag{16}$$

Apportioning this time to each low-latency packet, we can obtain the average forwarding delay for low-latency flows as

$$Latency_L = \frac{T_L}{F_L} = \frac{\sum_{i=1}^{N} R_{L_{i-1}} t_{l_i}}{R_{L_N}} = \frac{\sum_{i=1}^{N} (\prod_{j=0}^{i-1} \rho_j) t_{l_i}}{\prod_{i=0}^{N} \rho_i} \tag{17}$$

Similarly, we can make an estimate for non-low-latency flows:

$$T_{N_i} = F(\sum_{j=0}^{i-1} R_{P_j} t_{p_i} + R_{P_j} t_{n_i}) \tag{18}$$

$$T_N = F \sum_{i=1}^{N} [(1 - R_{L_{i-1}} t_{p_i}) + (R_{L_{i-1}} - R_{L_i}) t_{n_i}] \tag{19}$$

$$Latency_N = \frac{T_N}{F_N} = \frac{1}{1 - \prod_{i=0}^{N}} \sum_{i=1}^{N} [(1 - \prod_{j=0}^{i-1} \rho_j) t_{p_i} + (\prod_{j=0}^{i-1} \rho_j)(1 - \rho_i) t_{n_i}] \tag{20}$$

As a comparison, all traffic is processed by the pipeline without an FFP. The average packet forwarding delay can be denoted as

$$Latency = \sum_{i=1}^{N} t_{p_i} \tag{21}$$

Based on the above equations, we can summarize the benefits of low-latency traffic delay reduction:

$$
\begin{aligned}
W_L &= Latency - Latency_L \\
&= \sum_{i=1}^{N} t_{p_i} - \frac{\sum_{i=1}^{N}(\prod_{j=0}^{i-1} \rho_j) t_{d_i}}{\prod_{i=0}^{N} \rho_i} \\
&= \frac{1}{\prod_{i=0}^{N} \rho_i} [(\prod_{j=0}^{N} \rho_j) t_{p_i} - \sum_{i=1}^{N} (\prod_{j=0}^{i-1} \rho_j) t_{d_i}]
\end{aligned}
\tag{22}
$$

## 5. Experimental Results

To verify the feasibility of our approach, we conducted a series of experiments.

### 5.1. Simulation Setup

The emulator runs on a Linux platform with an Intel Xeon Silver 4208 CPU@2.10 GHz and 64 GB of RAM, and an Intel X710 is plugged as the NIC (Network Interface Card). The operating system is CentOS 7.9.2009. The software POF Switch is compiled using the O3 optimization option and works with DPDK 19.11.3 on the server. The ONOS controller works on another server with the same hardware configuration and connects to the POF Switch. Spirent SPTC50 is set as the traffic generator and the data analyzer. We use Spirent SPTC50 to generate traffic and input it into the POF Switch, and the switch processes the traffic and forwards it backward to the traffic generator. The forwarding latency and capability of the switch are collected by Spirent SPTC50 and shown in the Spirent TestCenter Application. The devices are connected as shown in Figure 9.

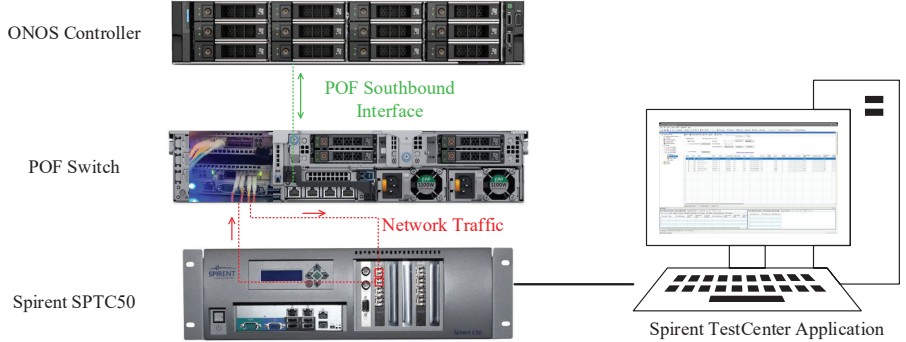

**Figure 9.** The setup of the experiments.

We simulated three real network scenarios:

- Communication in the big data processing. Communication between servers in DCNs requires basic routing and techniques such as an ACL to secure the network and servers. We use an ACL as a traffic processing step in our experiments to test the forwarding performance of the traffic.
- Videoconferencing. Videoconferencing is a typical use case of the publish–subscribe pattern in the present and future networks, with high requirements for video transmission quality and latency. We choose the SEADP (on-Site, Elastic, Autonomous Datagram Protocol) as the transport-layer protocol to encapsulate the traffic in our tests because it can provide ID-IP resolution-based routing and multiple QoS supports.
- IoV (Internet of Vehicles). As one of the core applications of 5G communication, the IoV demands an extremely low response latency. We use WSM (Wave Short Message) in our experiments to carry the simulated traffic of the IoV.

We conduct our experiments with a traffic processing model. The model consists of three processing stages. The first stage checks the Ethernet layer, including fields such as the destination MAC and Ethertype. Stage 1 distinguishes the messages encapsulated in the different network-layer protocols. The second stage includes the ACL table on the IPv6, the SEADP table, and the WSM table, which implement the QoS applications for the above three types of traffic, respectively. The third stage is an FIB (Forward Information Database), which consists of routing entries and matches only the destination IP or MAC.

Figure 10 shows the traffic processing flow. With the above configuration, the MAT in stage 1 and stage 2 are MM tables, while the FIB table matching the destination IP and the destination MAC in stage 3 are the LPM and EM, respectively.

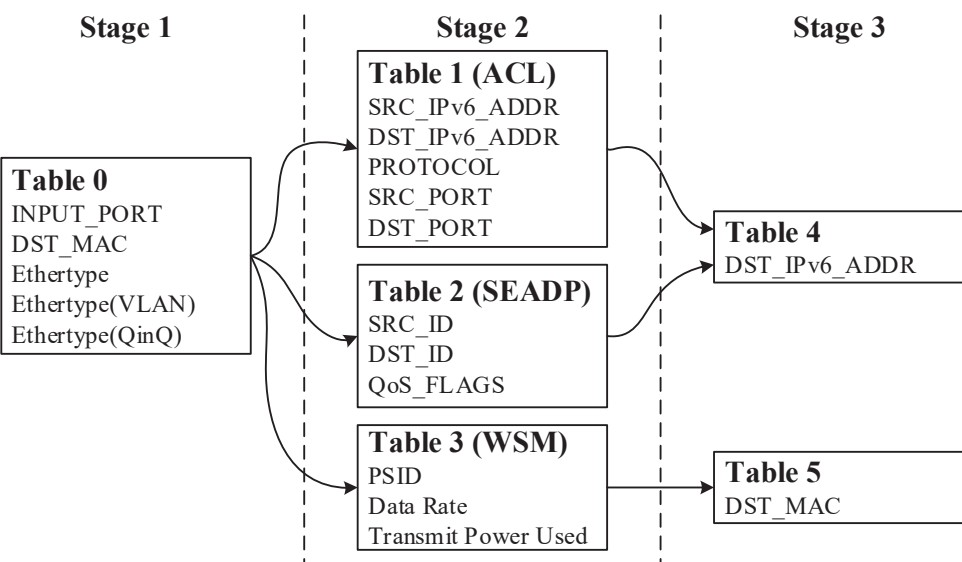

**Figure 10.** The traffic forwarding process.

First, we tested the forwarding performance of the simple pipeline POF Switch and the composite pipeline one proposed in this paper for low-latency flows. We measured the performance by the forwarding delay. We then used Spirent SPTC50 to generate mixed traffic containing low-latency flows to simulate real scenarios. We tested the improvement in the composite pipeline on the performance of the low-latency traffic forwarding and the cost of the increased latency of other flows forwarding in the mixed traffic scenario.

5.1.1. Forwarding Performance For Low-Latency Traffic

In this experiment, we used pure low-latency flows as the input traffic. For the native POF Switch, these flows are forwarded by the pipeline, while in our composite pipeline POF Switch, the processing of these flows is performed entirely in the FFP. We can use this test to learn the packet processing capabilities of the FFP and the pipeline, respectively. The average packet length is adjusted to test the maximum throughput and the forwarding delay without the packet loss in each scheme.

Figure 11 shows the forwarding capability of the switch when processing the traffic with different average packet lengths in the three scenarios. It is clear that when processing the server communication and videoconferencing traffic, the MAT pipeline in the native POF Switch cannot forward packets at the wire speed (10 Gbps) until the packets are longer than 640 bytes. In contrast, our proposed FFP has a better forwarding performance and can support wire-speed forwarding for 256-byte and longer packets. While for the forwarding capability of the IoV traffic, the MAT pipeline has a similar performance to our approach.

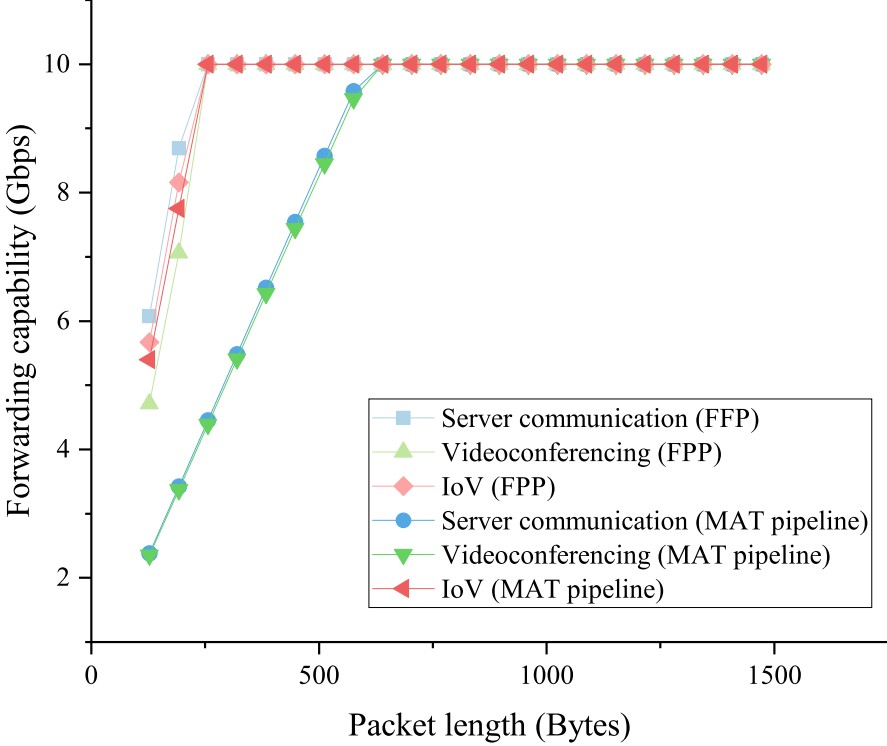

**Figure 11.** The low-latency traffic forwarding capabilities of the FFP and the MAT pipeline in the native POF Switch.

To make the results more convincing, we kept the packet length above 640 bytes so that both schemes can work at the wire speed. Then, we measured the forwarding latency, and the statistical results are depicted in Figure 12. It is easy to find that our proposed FFP has a much lower forwarding delay for server communication and videoconferencing traffic. By calculating the results, we can obtain a forwarding latency reduction of about 62–68%. As in the previous forwarding throughput test, the forwarding latency of the IoV traffic still does not show a significant difference between the two methods in this test. This is because the IoV traffic uses the efficient WSM protocol, in which case the difference in the search time between the hash lookup and MM table is not obvious due to the short input data.

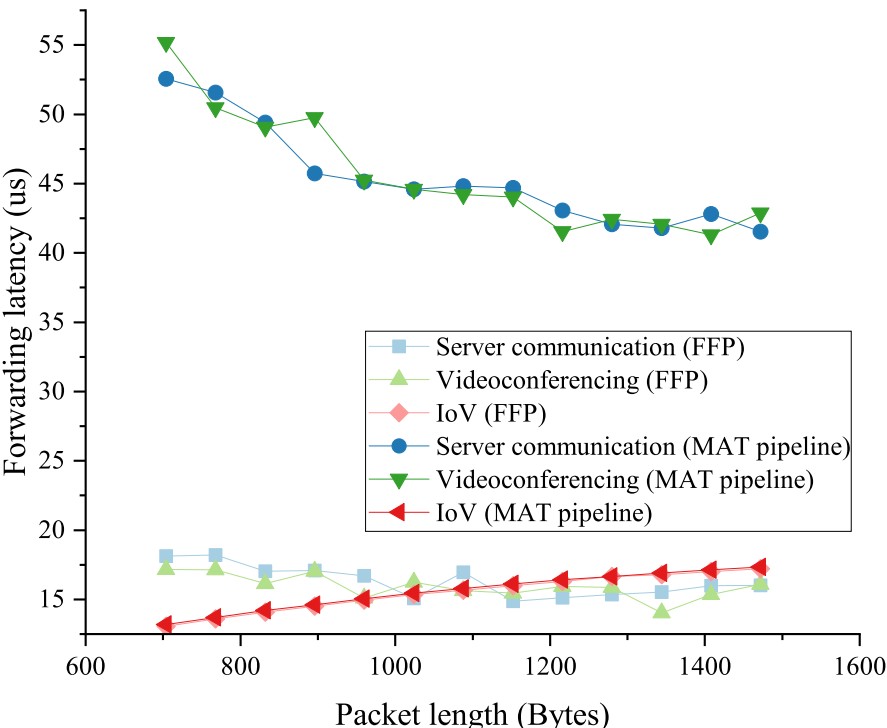

**Figure 12.** The low-latency traffic forwarding latency of the FFP and the MAT pipeline in the native POF Switch.

5.1.2. Forwarding Performance in Mixed Traffic Scenarios

Actual network traffic is composed complexly, so we used the traffic generator to generate mixed traffic to test the forwarding capability of the native POF Switch and the composite pipeline based on the above experiments. It should be noted that to work efficiently in real networks, our composite pipeline needs to ensure that the impact on normal traffic forwarding processes is within an acceptable range. For the above requirements, we tested the forwarding delay of low-latency flows and non-low-latency ones by adjusting the proportion of the low-latency part.

As Figures 13 and 14 show, for the server communication and videoconferencing part of the mixed traffic, our proposed scheme can complete the forwarding work with lower latency than the native POF Switch. In addition, for the non-low-latency part, the composite pipeline also has a better performance. The reason is that the FFP takes up part of the forwarding process for these flows, and the delay reduction is more than the delay increase caused by the packet transmission between the FFP and the MAT pipeline. Crucially, the above conclusion holds, regardless of the percentage of the low-latency portion of the mixed traffic.

However, for the IoV flows, as analyzed in the previous tests, the FFP cannot provide an effective optimization to the forwarding performance. Therefore, the delay reduction in the FFP is less than the delay increase in the packet transmission, and the forwarding latency shown in Figure 15 can be explained. All these results are consistent with the conclusions we analyzed in the previous sections.

Based on the above experimental data, we compiled the following table listing the round-trip time and reliability of different flows at a fixed percentage in the mixed traffic (20%). Table 3 shows the result of the comparison.

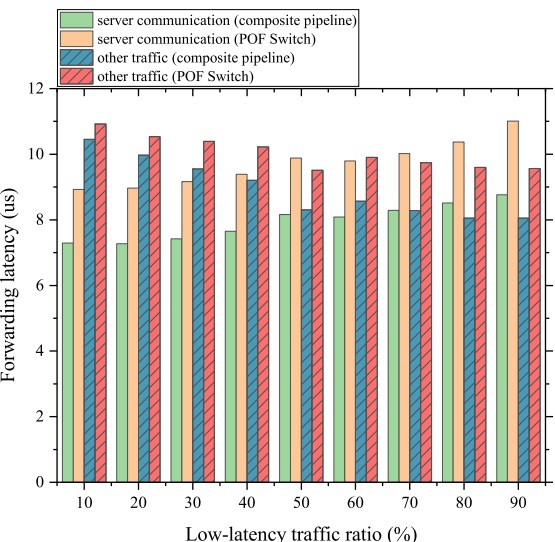

**Figure 13.** Forwarding latency of server communication traffic mixed with normal traffic.

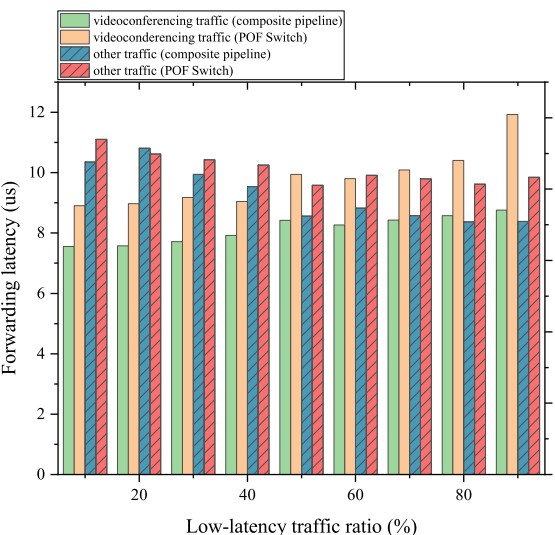

**Figure 14.** Forwarding latency of videoconferencing traffic mixed with normal traffic.

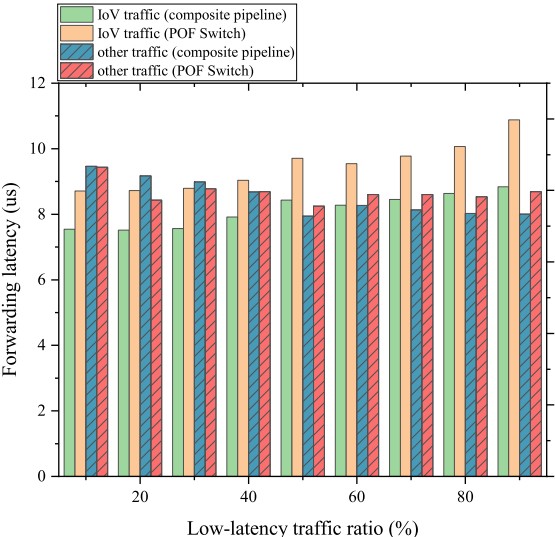

**Figure 15.** Forwarding latency of IoV traffic mixed with normal traffic.

**Table 3.** The average round-trip time and the reliability of traffic in the three scenarios.

| Traffic Type | Average Round-Trip Time (μs) | Reliability |
|---|---|---|
| Server communication | 14 ± 0.5 | 99.99999% |
| Videoconferencing | 15 ± 0.2 | 99.99999% |
| IoV | 15 ± 0.2 | 99.999% |

To show our approach achievements, we compared the round-trip time of the existing schemes and ours in Figure 16. The figure shows that due to the disability of distinguishing low-latency traffic, Approaches 1–5, respectively, perform an identical round-trip time for the normal/low-latency traffic. In contrast, our composite pipeline shows the disparity between the two types of traffic: the round-trip time for the low-latency traffic is much less, which meets the requirements of priority.

Table 4 lists the detailed data and the contribution that each approach makes to the reduction in the round-trip time. As shown in the table, some existing approaches greatly reduce the forwarding latency of the data-plane devices, such as PacketShader, by taking advantage of high-performance hardware. However, our approach is designed in software, and the result of the comparison shows the novelty of our approach—the obvious reduction in the round-trip time for the low-latency traffic.

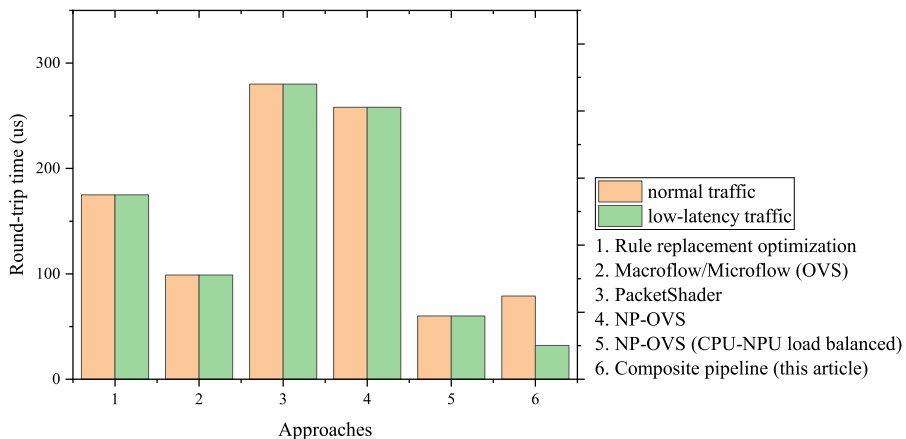

**Figure 16.** The round-trip time of normal/low-latency traffic in each approach.

**Table 4.** The average round-trip time and the reliability of traffic in the three scenarios.

| Approaches | Average Round-Trip Time before Optimization (μs) | Average Round-Trip Time after Optimization (μs) | Reduction for Normal Traffic | Reduction for Low-Latency Traffic |
|---|---|---|---|---|
| Rule replacement optimization | 184 | 175 | 4.89% | - |
| Macroflow/ Microflow (OVS) | 172 | 99 | 42.44% | - |
| PacketShader | >1000 | 280 | >72.00% | - |
| NP-OVS | 313 | 258 | 17.57% | - |
| NP-OVS (CPU-NPU load balanced) | 234 | 60 | 74.36% | - |
| Composite pipeline (this article) | 90 | 32 | <10% | 64.44% |

## 6. Conclusions

Our approach aims to improve the forwarding efficiency of low-latency traffic on SDN programmable data planes. In this paper, we provide the following contributions. Firstly, we proposed an identifying method for low-latency traffic based on the POF southbound interface protocol, which is accomplished jointly by marking the MAT entries on the control plane and labeling the packets on the data plane. Then, based on the above work, we proposed a composite pipeline architecture to improve the forwarding efficiency of low-latency traffic by introducing an FFP, in which we also made special optimizations to LPM tables. Finally, we demonstrated the effectiveness of our scheme through a series of experiments. The results show that the low-latency traffic forwarding capability of the FFP in the composite pipeline is much better than that of the MAT pipeline. Specifically, in DCN server communication and videoconferencing scenarios, we reduced the lower limit of the average packet length required for wire-speed forwarding from 640 to 256 bytes, and the forwarding capacity of the FFP is more than twice that of the MAT pipeline when processing packets shorter than 256 bytes. In terms of forwarding latency, we tested wire-speed input traffic. The results showed that the forwarding delay of low-latency flows is reduced by more than 60%. In addition, we also measured the forwarding capability of mixed traffic with different percentages of the low-latency part. The results show that the composite pipeline reduces the forwarding delay of the low-latency flows by 17–20%, and the forwarding delay of the non-low-latency part has a similar reduction. While in the simulated IoV scenario, the forwarding process consists of efficient MATs with less searching time, the improvement in our approach is not outstanding enough, and the forwarding latency even increases slightly. To sum up, our work in this paper has a certain degree of improvement for low-latency traffic forwarding in SDN data planes.

In order to make our approach more efficient, our future work will focus on optimizing the FFP, including improving the lookup hit rate of LUTs, reducing the lookup time for short matching fields. When the optimization completes, we will further offload the FFP module to hardware devices such as FPGAs with a higher performance.

**Author Contributions:** Conceptualization, Z.L., X.C. and L.S.; methodology, Z.L., X.C. and L.S.; software, Z.L.; validation, Z.L., X.C. and L.S.; writing—original draft preparation, Z.L.; writing—review and editing, X.C. and L.S.; visualization, Z.L.; supervision, X.C. and L.S.; project administration, L.S.; funding acquisition, X.C. All authors have read and agreed to the published version of the manuscript.

**Funding:** This work was funded by the Strategic Leadership Project of Chinese Academy of Sciences: SEANET Technology Standardization Research System Development (Project No. XDC02070100).

**Institutional Review Board Statement:** Not applicable.

**Informed Consent Statement:** Not applicable.

**Data Availability Statement:** Not applicable.

**Acknowledgments:** We would like to express our gratitude to the reviewers for their helpful comments.

**Conflicts of Interest:** The authors declare no conflict of interest.

## Abbreviations

The following abbreviations are used in this manuscript:

| | |
|---|---|
| ICN | Information-Centric Networking |
| DCN | Data Center Networking |
| SDN | Software-Defined Networking |
| QoS | Quality of Service |
| MAT | Match-Action Table |
| POF | Protocol-Oblivious Forwarding |
| FFP | Fast-Forwarding Path |

| | |
|---|---|
| PISA | Protocol-Independent Switch Architecture |
| TCAM | Ternary Content Addressable Memory |
| OVS | Open vSwitch |
| NP | Network Processor |
| FRT | Forwarding Rule Table |
| LUT | LookUp Table |
| EM | Exact Matching |
| LPM | Longest Prefix Matching |
| MM | Mask Matching |
| NDN | Named Data Networking |
| ACL | Access Control List |
| SEADP | on-Site, Elastic, Autonomous Datagram Protocol |
| IoV | Internet of Vehicles |
| WSM | Wave Short Message |
| FIB | Forwarding Information dataBase |

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
