# Peer review of "A Composite Pipeline for Forwarding Low-Latency Traffic in SDN Programmable Data Planes"

_electronics, doi:10.3390/electronics12020461_

Round 1
Reviewer 1 Report
Comments to authors:
The paper needs improvement in English.
Check your appreciation. What it stands for “MAT”, “POF”? The authors should have a table explaining the terminology introduced so the reader can check this at any time.
The novelty of your approach is not evident in your introduction; show the novelty clearly.
Clearly state the contributions in a list in the introduction section.
The motivation should go to the “Introduction” section.
Rename the section “Background and Motivation” to “Literal Review and Background Work” and move the contents in the paper (e.g., introduction) related to background work to the subsection of “Background Work”. Also, move the section “Related Work” to this section.
Make a comparison table that you show clearly your novelty by comparing the existing approaches in the Section “Literal Review and Background Work”.
Add a section “Problem Description” that you need to describe the problem.
In the section “The Proposed Mechanisms” add a diagram that shows the complete system with network equipment, servers and clients.
In the section” Experimental Results” add your assumptions, and system setup and explain how you implement the simulation and emulation (tools, simulation tools, libraries). For example, you used mininet with vswitches.
Compare your approach under the "Experiment and Results analysis" section with other approaches found in the literature (from your related work section) and show your approach achievements using Figures (minimum five approaches). Please, in the comparison, add a table on the metrics you are comparing.
Explain in more detail and depth the results of Figures 9-10 and 11-13.
In your simulation and emulation consider the URLLC requirements and show with Figures the round trip time of a packet in order to show that the latency is reduced. Thus, in a Table add the probability or error and the latency time in μseconds.
In the "Experiment and Results analysis" compare your approach with others found in the literature review, using similar metrics shown in the literature review (e.g., latency time)
Add your future work plans in the "Conclusions" section and add the comparison results with approaches from Related Works (as instructed) here.

Reviewer 2 Report
The paper proposed a composite pipeline that processes low-latency flows in a more efficient path. The strong point is the design of the proposed mechanisms. The experiment results are concluded by forwarding latency metrics. However, the following points are addressed to the authors:
1. The evaluation of the processing latency in the pipeline in the proposed mechanism is lacking.
2. How does the forwarding latency of videoconferencing traffic mixed with normal traffic behave with the length of the pipeline?
Reviewer 3 Report
Strong aspects:
The solution is interesting.
The ideas are clearly presented.
The English is good.
The proposal is validated through experiments.
A few typos, e.g.:
On page 4, in "as shown in Fig 1,." the comma must be removed.
Some remarks regarding the paper:
It is not very clear the setup for the experimental part.
It seems that the traffic forwarding process is emulated, both for the native POF switch and for the composite pipeline, and different type of traffic is fed to the forwarding process and the forwarding latency is estimated.
Is it so? The section explaining the experimental setup could be improved.
The conclusions present only the positive aspects of the proposed solution. Does the solution have any drawbacks?
Round 2
Reviewer 1 Report
Comments to authors:
The paper needs minor improvements in English.
Compare your approach under the "Experiment and Results analysis" section with other approaches found in the literature (from your related work section) and show your approach achievements using Figures (minimum five approaches) by adding a table that shows your novelties over the other. In the comparison above table, add the metrics you are using vs the other approaches metrics as a column.
In the "Experiment and Results analysis", compare your approach with others found in the literature review, using similar metrics shown in the literature review (e.g., latency time) and make extra figures showing your superiority.

Author Response
We believe that your two comments convey the same suggestion, so we have added a figure (Fig 16 in Page 20) to compare the contribution of reducing the round-trip time for low-latency traffic that existing approaches and ours makes. And a table (Table 4 in Page 20) is added as well to list detailed experimental data, and it is easy to find that our approach greatly reduces the round-trip time for low-latency traffic, which is our novelty. In addition, the explanation of the figure and the table is inserted at the end of the section “Experimental Results” (Line 590-601 in Page 19).

Reviewer 2 Report
The authors have addressed all my comments.
Author Response
Thank you again for your previous comments and your recognition of our work.
Round 3
Reviewer 1 Report
Comments to authors:
The paper needs minor improvements in English.
All requested improvements are implemented in the manuscript, and the document's quality is enhanced as a result of the paper meeting publication standards.